# Structure, Vibrational Spectra, and Cryogenic MatrixPhotochemistry of 6-Bromopyridine-2-carbaldehyde: From the Single Molecule of the Compound to the Neat Crystalline Material

**DOI:** 10.3390/molecules28041673

**Published:** 2023-02-09

**Authors:** Anna Luiza B. Brito, Susy Lopes, Gulce Ogruc Ildiz, Rui Fausto

**Affiliations:** 1CQC-IMS, Department of Chemistry, University of Coimbra, 3004-535 Coimbra, Portugal; 2Department of Physics, Faculty of Science and Letters, Istanbul Kultur University, Atakoy Campus, Bakirkoy, Istanbul 34156, Turkey

**Keywords:** 6-Bromopyridine-2-carbaldehyde, matrix isolation, UV–vis, infrared, Raman and mass spectra, photochemistry, DFT calculations, Hirshfeld analysis

## Abstract

6-Bromopyridine-2-carbaldehyde (abbreviated as BPCA) is used both as a building block in supramolecular chemistry and as a ligand for transition metal catalysts and luminescent complexes. In this study, the structure and vibrational spectra of BPCA were investigated in both the room temperature neat crystalline phase and for the compound isolated in cryogenic Ar, Kr and Xe matrices. The experimental studies were complemented by quantum chemical DFT(B3LYP)/6-311++G(d,p) calculations. For the crystalline compound, infrared and Raman spectra were obtained and interpreted. Comparison of the obtained infrared spectrum of the crystal with those obtained for the isolated molecules of BPCA in the studied cryomatrices helped to conclude that the intermolecular interactions in the crystal do not significantly perturb the intramolecular vibrational potential. Structural analysis further supports the existence of weak coupling between the intermolecular interactions and the structure of the constituting molecular units in crystalline state. The intermolecular interactions in the BPCA crystal were also evaluated by means of Hirshfeld analysis, which revealed that the most important interactions are weak and of the H^…^N, H^…^O, H^…^H, H^…^Br and Br^…^Br types. The conformer of BPCA present in the crystal was found to correspond to the most stable form of the isolated molecule (*trans*), which bears stabilizing C–H^…^O=C and C(=O)H^…^N interactions. This conformer was shown to be the single conformer present in the as-deposited cryogenic matrices prepared from the room temperature gaseous compound. Broadband UV irradiation of matrix-isolated BPCA (λ ≥ 235 nm) resulted in the conversion of the *trans* conformer into the higher-energy *cis* conformer, where repulsive C–H^…^H–C(=O) and C=O_LP_^…^_LP_N (where LP designates a lone electron pair) interactions are present, and decarbonylation of the compound with formation of 2-bromopyridine (plus CO). The decarbonylation reaction was found to be more efficient in the more polarizable Xe matrix, indicating stabilization of the radicals initially formed upon breaking of the C–C(HO) and C–H (aldehyde) bonds in this medium, and testifying the occurrence of the decarbonylation reaction with involvement of radical species. TD-DFT calculations were used to access the nature of the excited states associated with the observed UV-induced reactions. As a whole, this study provides fundamental data to understand the physicochemical behavior of the compound, bridging the properties of the isolated molecule to those of the neat crystalline com-pound. Such information is of fundamental importance for the understanding of the role of BPCA in supramolecular chemistry and to potentiate its applications in synthesis and as a ligand for transition metal catalysts and luminescent complexes.

## 1. Introduction

6-Bromopyridine-2-carbaldehyde (abbreviated as BPCA) is used both as a building block in supramolecular chemistry and as a ligand for transition metal catalysts and luminescent complexes [1,2]. The compound has also been used in the synthesis of porphyrins [3].

From the structural point of view, BPCA appears as an interesting compound due to the following characteristics: *(i)* the molecule of the compound possesses an internal rotational degree of freedom (the torsion around the exocyclic C–C bond), so that different conformers can be expected to exist; *(ii)* the location of the aldehyde substituent at the *ortho* position to the pyridine nitrogen atom can be anticipated to lead to relevant conformation-dependent intramolecular interactions between these two moieties; *(iii)* the presence of the voluminous, strongly polarizable bromine substituent in the molecule opens expectations regarding its importance in determining fundamental characteristics of the crystal structure of the compound; *(iv)* the unimolecular photochemistry of the isolated molecules of the compound appears as an interesting topic to be investigated, considering the *a priori* possibility of occurrence of both UV-induced conformational isomerization [4,5,6] and photofragmentation reactions, specifically decarbonylation [6,7,8], pyridine ring valence isomerization [9,10,11], and Br atom extrusion [12,13]. The investigation of all these characteristics appears as a fundamental step for the detailed understanding of the physicochemical behavior of BPCA, bridging the properties of the isolated molecule to those of the neat crystalline compound and potentiating its use in supramolecular chemistry, in synthesis, and other practical applications, as those mentioned above.

The crystalline structure of BPCA is known [14], the crystals being monoclinic, space group *P*2_1_/*a* (no. 14), with *a* = 6.908(2) Å, *b* = 6.290(4) Å, *c* = 15.060(6) Å, and β = 95.57(3)°, at 290 K, and four molecules per unit cell (*Z* = 4). In the crystal, the molecules are essentially planar (with an r.m.s. deviation of 0.006(4) Å for non-H atoms) and assume a conformation where the oxygen aldehyde atom points to the opposite direction of the nitrogen ring-atom (*trans* form). Molecules are linked into chains along the direction of the crystallographic *b* axis by weak intermolecular C–H^...^N hydrogen-bond-like interactions (d_H…N_ = 2.53(6) Å; d_C…N_ = 3.534(7) Å; ∠_C–H…N_ = 167(4)°) and form columns along the *a* axis direction, where the molecules are symmetry related by a 2_1_ screw-axis symmetry transformation, so that in adjacent layers, they are rotated by ca. 180° in relation to each other, and the nitrogen atom of a given molecule stays nearly above the center of the rings of the neighboring molecules. When viewed along the direction of the *b* axis, the columns formed by the stacked BPCA molecules are separated by alternate regions containing C–H^...^O and C–H^...^Br contacts (d_H…O_ = 2.64(6) Å; d_C…O_ = 3.482(7) Å; ∠_C–H…O_ = 138(4)°; d_H…Br_ = 3.24(6) Å; d_C…Br_ = 4.033(6) Å; ∠_C–H…Br_ = 148(5)°) (see Appendix A) [14]. To the best of our knowledge, no other structural, vibrational or photochemical studies (either experimental or theoretical) have been reported on BPCA, although its mass spectrum (MS) and room temperature infrared (IR) spectrum are available in SpectraBase (spectrum ID: zJEhEVy4MC and 2XLMvrNG6Oo, respectively) [15].

In the present study, in order to structurally and vibrationally characterize BPCA in detail, and in particular in aiming to bridge the properties of the isolated molecule to those of the neat crystalline compound, BPCA was investigated in both the room temperature neat crystalline phase and isolated in cryogenic Ar, Kr and Xe matrices. The experimental studies were complemented by quantum chemical DFT(B3LYP)/6-311++G(d,p) calculations. For the crystalline compound, infrared and Raman spectra were obtained and interpreted, and the most important intermolecular interactions were evaluated by means of Hirshfeld analysis (which was performed using the X-ray diffraction structural data previously reported by Zhang and coworkers) [14]. The matrix-isolated molecules of the compound were also characterized structurally and vibrationally using infrared spectroscopy, and the results of the irradiation of the matrix-isolated molecules with broadband UV light (λ ≥ 235 nm) were evaluated. As described in detail below, both UV-induced conformational isomerization and decarbonylation of the compound, with formation of 2-bromopyridine (plus CO), were observed.

## 2. Experimental Methods

6-Bromopyridine-2-carbaldehyde was obtained commercially (purity > 97%) and used as purchased. For the study of the isolated molecules of the compound, the matrices were prepared by co-deposition of the sublimed BPCA and Ar, Kr or Xe (purity: N60, N48 and N48, respectively; all gases being obtained from Air Liquide) onto a CsI substrate assembled at the cold (15 K) tip of an APD Cryogenics DE-202A closed-cycle He-refrigerated cryostat. Absence of significant aggregates of the compound in the matrices was confirmed by inspection of the IR spectra, which do not show bands ascribable to these species. The compound was sublimed using a home-made variable temperature Knudsen cell, which was connected to the cryostat through a needle valve, both the cell and the valve nozzle being kept at room temperature (25 °C = 298.15 K) during deposition of the matrices. The temperature of the cold CsI window was measured at the sample holder using a silicon diode sensor and a temperature controller LakeShore 335 (accuracy: 0.1 K).

The matrix isolation IR spectra (in the 4000–400 cm^−1^ spectral range) were obtained using a Thermo Nicolet 6700 FTIR spectrometer, with 0.5 cm^−1^ spectral resolution. The spectrometer is equipped with a mercury cadmium telluride (MCT-B) detector and a Ge/KBr beam splitter, and was continuously purged by a stream of dry, CO_2_-filtered air during the experiments in order to minimize interference of atmospheric H_2_O and CO_2_. The IR spectrum of the solid crystalline compound was obtained in the attenuated total reflection mode (ATR) using a Thermo Scientific FTIR Nicolet iS5 system, with an iD7 ATR accessory (angle of incidence: 45°; crystal: diamond). The spectrum was recorded with spectral resolution of 2 cm^−1^, in the wavenumber range of 4000–400 cm^−1^, being the average of 64 scans. Raman data were obtained for the powder of the compound (after fine grinding of the crystalline material), at room temperature, in a Raman Horiba LabRam HR Evolution system, with excitation at *λ* = 532 nm. The spot diameter of the laser was approximately 1 µm and was focused on the sample through a 50× long-working-distance objective (numerical aperture 0.5). A laser power of ~5 mW (at the sample) was used, the final spectra being the average of 50 accumulations of individual spectra collected during 10 s, with the spectral resolution of 2.0 cm^−1^. The calibration of the system was performed using as reference the silicon crystal Raman peak at 520.5 cm^−1^. The UV–visible (UV–vis) spectrum of the compound (in ethanol solution) was recorded, in the range of 170–800 nm, at room temperature, in a 10 mm path length high-precision quartz cell, using a ScanSpec UV–vis ScanSci spectrometer, with an auxiliary Hamamatsu light source (model L10761) (integration time per point: 4 s; counts: 50).

UV irradiations of the matrices were carried out through an external KBr window of the cryostat (cut-off wavelength: 235 nm) using a high-pressure 200 W Hg(Xe) lamp (Newport, Oriel Instruments), the infrared component of the light beam being filtered by means of a water filter (path length: 9.5 cm).

## 3. Computational Details

The Density Functional Theory (DFT) calculations were performed with Gaussian 09 (A.02 revision) [16] using the B3LYP functional [17,18,19] and the 6-311++G(d,p) basis set [20,21,22], the optimized structures of the stationary points described in this study being confirmed to correspond to a true minimum or a first-order transition state by analysis of the corresponding Hessian matrix.

Calculated harmonic vibrational frequencies and intensities (obtained at the same level of theory) were used to simulate the spectra presented in the figures, by convolution with Lorentzian functions (full width at half maximum: 1.5 cm^−1^). The calculated wavenumbers were scaled down by 0.983 and 0.955, below and above 1800 cm^−1^, respectively, to account for the incomplete nature of the basis set, approximated theoretical method and, mostly, anharmonicity. Assignments were made with help of normal coordinate analysis calculations following the procedure first described by Schachtschneider and Mortimer [23], the chosen internal coordinates being defined as recommended by Pulay et al. [24] (see Appendix A).

Time-dependent DFT (TD-DFT) calculations were carried out using with the long-range corrected CAM-B3LYP [25] Coulomb attenuated function (which has been shown to provide accurate vertical excitation energies for a wide variety of organic molecules) [26], together with the 6-311++G(d,p) basis set. The bulk solvent effects of ethanol were considered within the polarizable continuum model (PCM) framework, using the integral equation formalism variant (IEFPCM) [27,28].

Hirshfeld surface analysis [29,30] was undertaken using the Crystal Explorer 17.5 software [31], with the structure input file of BPCA obtained in the CIF format [14].

## 4. Results and Discussion

### 4.1. DFT Calculations: Conformers, Relative Energies and Barriers for Conformational Isomerization

The molecule of BPCA has a single conformationally relevant internal degree of freedom, which is the rotation around the exocyclic C–C bond connecting the aldehyde group to the aromatic ring. Two *C*_s_ symmetry planar conformers (*cis* and *trans* forms, represented in Figure 1) were found on the potential energy surface of the molecule. The conformers differ in energy by 17.7 kJ mol^−1^, with the *trans* form being the most stable conformer (17.1 kJ mol^−1^, after consideration of the zero-point correction; ΔG°_(298.15 K)_ = 16.7 kJ mol^−1^), and are separated by an energy barrier of 21.0 kJ mol^−1^ (19.3 kJ mol^−1^, upon consideration of the zero-point correction; ΔG°^#^_(298.15 K)_ = 21.0 kJ mol^−1^) in the *cis*→*trans* direction (Figure 1). The estimated room temperature gas-phase equilibrium populations of the conformers were estimated from the calculated relative Gibbs energies (ΔG°_(298.15 K)_) using the Boltzmann equation, and amount to 99.9% (*trans*) and 0.1% (*cis*), i.e., in these conditions, only the most stable *trans* conformer is significantly populated.

The relative energy of the two conformers is mostly dictated by interactions between the aldehyde group and both the nitrogen and hydrogen atoms’ *ortho* to this substituent, which are stabilizing in the most stable *trans* conformer (being of the C–H^…^O=C and C(=O)–H^…^N types), and are repulsive in the higher-energy *cis* form (C–H^…^H–C(=O) and C=O_LP_^…^_LP_N interactions, where LP designates a lone electron pair). The attractive interactions in the *trans* conformer shall not be classified as intramolecular hydrogen bonds, since they do not satisfy the geometric criteria for this type of interaction [32,33], with the C–H7^…^O and C(=O)H^…^N angles being exceedingly small (93.5° and 70.9°, respectively). These interactions are better described as attractive bond–dipole/bond–dipole contacts, established between the pairs of nearly anti-parallel, charge-polarized bonds [^(−δ)^C–H7^(+δ)^/^(−δ)^O=C^(+δ)^], and [^(−δ)^C(=O)–H^(+δ)^/^(−δ)^N–C2^(+δ)^]. The repulsive interactions in the less stable *cis* conformer also have a bond–dipole/bond–dipole component; in this case, the dipoles associated with the interacting bonds are aligned in an approximately parallel fashion.

Table 1 presents some relevant structural parameters calculated for the two conformers of BPCA. This table also shows the corresponding experimental values measured in the crystal of the compound by X-ray diffraction (XRD) [14], for comparison. The full calculated optimized geometries of the conformers (Cartesian coordinates) are provided as Appendix A.

The distinct intramolecular interactions present in the conformers of BPCA lead to different geometrical parameters of the aldehyde group and its vicinity, while the C–Br bond length is predicted by the calculations to be nearly identical in both forms (1.924 and 1.925 Å, in conformers *trans* and *cis*, respectively). The exocyclic C2–C11 bond is shorter in the most stable *trans* conformer than in the *cis* form, this being partially determined by an increased π delocalization involving the aldehyde fragment and the aromatic ring in the *trans* conformer, and partially due to the augmented strain resulting from the repulsive interactions that operate in the *cis* conformer in comparison with the attractive interactions that are present in the *trans* form that have been mentioned above. Being the strongest repulsive interaction present in the *cis* conformer, the C=O_LP_^…^_LP_N interaction reflects itself also in the considerably larger N–C2–C11 and C–C=O angles in this conformer (118.1 and 125.3°, respectively), compared to the *trans* form (115.3 and 123.7°). The increase in the N–C2–C11 and C–C=O angles in the *cis* conformer is mostly compensated by the reduction of the C3–C2–C11 and O=C–H angles in going from the *trans* to the *cis* form, changing from 121.6 to 119.2° and from 122.5 to 121.2°, respectively. On the other hand, in accordance with a more important π electron delocalization from the aromatic ring to the aldehyde group, the calculated C=O bond length is longer in the *trans* conformer than in the *cis* form (1.208 vs. 1.203 Å).

In general, the calculated geometrical parameters for the *trans* conformer, which is also the form present in the crystal of the compound [14], do not differ considerably from the XRD experimental values, although the ring CC bonds and the C–Br bond have been predicted somewhat longer than observed. Noteworthy, the calculated angles fit very well to the experimental ones. Since the calculations were performed for the isolated molecule and the geometrical parameters are not affected by vibrational contributions, while the experimental values correspond to molecules in a crystal, subjected to intermolecular interactions, and the data are affected by vibrations, such a good correspondence between the calculated and experimentally determined geometrical parameters is an indication of the relatively small perturbations imposed by the intermolecular interactions present in the crystal of BPCA on the intramolecular potential. This indication is validated by both the spectroscopic and Hirshfeld analyses of crystalline BPCA, as it will be described in detail in Section 4.4.

An additional note shall be here made in relation to the energy barrier for conformational isomerization. In the *cis*→*trans* direction, this barrier was predicted to be 21.0 kJ mol^−1^ (see Figure 1), confirming that both conformers are located in well-defined minima and that at the low temperatures typical of a matrix isolation experiment (10–15 K) they cannot interconvert via the over-the-barrier process. In addition, to take place, the conformational isomerization has to involve a substantial movement of the aldehyde oxygen atom, so that the calculated isomerization barrier is certainly high enough to prevent spontaneous conversion of the *cis* conformer into the *trans* form via quantum mechanical tunneling. This means that, if the *cis* conformer can be generated in situ by any method, it can be expected to be stable under low temperature matrix isolation conditions, since both the over-the-barrier and tunneling-assisted processes are not accessible. As described in Section 4.3, this expectation was confirmed in the present study, with the *cis* conformer being produced by in situ UV excitation of the *trans* conformer initially trapped into the cryogenic matrices from the gas phase of the compound.

### 4.2. Infrared Spectra of Matrix-Isolated BPCA

Molecules of BPCA were isolated in Ar, Kr and Xe matrices as described in Section 2, the obtained spectra being presented in Figure 2, where they can be compared with the simulated IR spectrum of the *trans* conformer built using the B3LYP/6-311++G(d,p) calculated vibrational data (see Section 3 for details).

The calculated spectrum of the *trans* conformer fits very well the experimental spectra, indicating that only this form is present in the as-deposited matrices. The proposed assignments are provided in Table 2, together with the calculated potential energy distribution (PED) obtained from the performed normal coordinate analysis. Considering the very good agreement between the experimental and calculated data, the assignments are straightforward, and the following discussion focuses only on a few spectral features that deserve some additional comments.

In the high-frequency region, the bands due to the three CH stretching ring modes are observed in the 3130–3045 cm^−1^ range, and those originating in the aldehyde C–H stretching mode are seen between 2860 and 2810 cm^−1^. As usually, the C–H stretching aldehyde mode gives rise to a complex-structured band appearing at considerably low frequencies. The multiplet profile of the band observed in all studied matrices results from matrix-site splitting (different trapping sites for the molecules) and Fermi resonance due to coupling with the first overtone of a mixed CH ring in-plane bending and ring stretching mode whose fundamental is observed in the 1440–1416 cm^−1^ range (calculated value: 1431 cm^−1^). The low frequency of the aldehyde C–H stretching mode results from the weakening of the C–H aldehyde bond due to the back-donation effect from the lone electron pair of the carbonyl oxygen atom in position *trans* to the bond [34,35]. Matrix-site splitting is also observed in all matrices for most of the other observed bands, and Fermi resonances also account for the complex structure exhibited by some of other bands, as it will be highlighted below. The CH stretching ring modes are of low intensity but are clearly identifiable, and their frequencies follow the usual pattern in the different matrices, decreasing going from the Ar to the Kr to the Xe matrix. It is well known that for this type of vibration, an increased polarizability of the matrix host material (Ar < Kr < Xe) slightly changes the intramolecular vibrational potential resulting in smaller force constants for the vibrational oscillators and, consequently, in a shift to lower frequency values [36].

As for the CH aldehyde stretching mode, the C=O stretching vibration (predicted at 1751 cm^−1^) also gives rise to a complex band in all matrices, showing matrix-site splitting features coupled to Fermi resonance, the latter noticeable by the appearance of two major groups of bands in the 1740–1715 and 1715–1700 cm^−1^ frequency ranges. The Fermi doublet results from the interaction of the C=O stretching fundamental with the first overtone of a vibration with major contribution of the exocyclic C–C stretching coordinate, whose fundamental is also observed as a multi-maxima feature with main maximum at around 860 cm^−1^ (predicted at 849 cm^−1^). By looking to the calculated vs. experimental spectra represented in Figure 1, it appears at first sight that the intensity of the C=O stretching band is considerably overestimated by the calculations. However, this is not the case, since the total intensity behind the complex-band profile ascribable to this mode spawns for a wide range of frequency, and it is the highest of all bands appearing in the spectrum of BPCA in all investigated matrices.

Another vibration giving rise to a clearly visible Fermi doublet, split by matrix-site effects, is the ring stretching mode, mainly localized in the C4–C5 bond and the C6–N–C6 fragment (νring2), which has component-bands in the ~1600–1570 and 1570–1550 cm^−1^ ranges (predicted at 1569 cm^−1^). According to the calculations, also contributing to the total intensity of these spectral features is the ring stretching mode (νring1) that is essentially localized in the C2–C3 and C5–C6 bonds and is predicted at 1578 cm^−1^, as a low IR intensity vibration (see Table 2).

The assignment of the remaining bands is straightforward, since they are all very well predicted by the calculations (see Table 2 and Figure 1). The in-plane and out-of-the-plane CH aldehyde bending modes are observed in the 1350–1340 and 1010–1000 cm^−1^ ranges (calculated values: 1346 and 1011 cm^−1^), while the exocyclic C–C coordinate contributes significantly to the vibration associated with the band observed at ca. 475–470 cm^−1^ (predicted 468 cm^−1^) as well as to the mode related with the bands observed in the 1225–1205 cm^−1^ range (predicted at 1220 cm^−1^, and appearing as a Fermi doublet via interaction with the first overtone of the in-plane ring bending mode (δring2) whose fundamental is observed around 630 cm^−1^; predicted at 634 cm^−1^), besides to the above mentioned band observed at around 860 cm^–1^. The C–Br stretching coordinate contributes significantly to the band-doublet observed around 710 cm^−1^ (calculated 706 cm^−1^), which is the mode with a frequency above 400 cm^−1^ that presents the largest PED contribution of the C–Br stretching coordinate (the mode with highest contribution from the C–Br stretching coordinate has a calculated frequency of 297 cm^−1^, which lies below the range of frequencies investigated).

The aldehyde torsional mode is predicted at 95 cm^−1^, as the lowest frequency vibration, lying in a spectral region not accessible to our infrared spectroscopy experiments. In the Raman spectrum of the crystalline BPCA, a band is observed at 79 cm^−1^ (see Section 4.4) that seems to be the best candidate for this torsional vibration (there is also a band in the Raman spectrum observed at 93 cm^−1^, but with an intensity much lower than the predicted one, and that is probably due to an intermolecular mode).

### 4.3. UV-Induced Rotamerization in Matrix-Isolated BPCA

The matrix-isolated BPCA was subjected to UV radiation (λ ≥ 235 nm), as described in Section 2. The UV irradiation led to consumption of the initially deposited *trans* conformer and emergence of new bands in the spectra. As shown in Figure 3, the performed irradiation promoted both *trans*→*cis* BPCA conformational isomerization and decarbonylation (with production of 2-bromopyridine and CO). In the figure, the experimental IR difference spectrum obtained in the Xe matrix (irradiated matrix *minus* as-deposited matrix) is compared with the simulated IR difference spectrum built using the B3LYP/6-311++G(d,p) calculated spectra of the two BPCA conformers, and that of 2-bromopyridine. The results obtained in the other studied matrices (Ar and Kr) were similar, but the reactions were found to be less efficient. Under the irradiation conditions used (see Section 2), the photo-induced processes showed a low efficiency; in the most favorable case of the Xe matrix experiments, the total consumption of the reactant species was only about 5% after 180 min. of irradiation, as determined by comparing the relative intensities of the bands of the *trans* BPCA conformer before and after irradiation.

The assignment of the new bands originated in the *cis* conformer of BPCA are given in Table 3, together with the results of the normal coordinate analysis performed for this form. As it can be seen, from the 19 modes of conformer *cis* predicted to give rise to intense or medium intensity (≥5 km mol^−1^) infrared bands above 400 cm^−1^, 17 were observed experimentally after UV irradiation, at frequencies matching very well the calculated ones. Once formed, the *cis* conformer was found to be stable in the matrices, as predicted (see discussion in Section 4.1).

In the *cis* conformer, the CH stretching mode of the aldehyde group is observed at a considerably lower frequency compared to that of the *trans* form (2732 vs. 2847–2818 cm^−1^, respectively, in the Xe matrix), and it appears as a single band, indicating that in the *cis* form, the aldehyde CH stretching vibration is not participating in any Fermi resonance interaction (the data also show that matrix-site splitting is less significant for the photo-generated *cis* conformer). Because the aldehyde CH stretching vibration is shifted to a much lower frequency, the Fermi resonance that is observed in *trans* BPCA is removed in the *cis* form, since the frequency of the coupled mode is nearly the same in the two conformers (1436–1416 cm^−1^ in the *trans* conformer, and 1427 cm^−1^ in the *cis* form; Xe matrix data).

The occurrence of the decarbonylation reaction is clearly seen by the appearance of the characteristic band of carbon monoxide at 2133 cm^−1^ [37], while mark-bands of 2-bromopyridine are observed at 1577, 1451, 1417, 1113/1109, 1080, 1044, 760 and 703 cm^−1^ (Xe matrix data), matching well those predicted by the calculations for this compound at 1581/1577, 1455, 1420, 1106, 1077, 1042, 760 and 701 cm^−1^, respectively. Assignments in all matrices are provided in Table 4.

An interesting observation is that among the bands emerging upon irradiation, those ascribed to 2-bromopyridine are considerably broader than those assigned to the *cis* BPCA conformer, a result that can be explained by the fact the photoproduced 2-bromopyridine molecules have to share the original matrix-cage with CO.

The spectra of the irradiated matrices do not show any evidence of occurrence of other photo-induced reactions, in particular pyridine ring valence isomerization [9,10,11] and Br atom extrusion [12,13].

The decarbonylation of aromatic aldehydes has been observed frequently as a result of UV irradiation of these type of compounds under matrix isolation conditions [6,7,8,38]. The fact that the reaction was found to be more efficient in the Xe matrix indicates stabilization of the radical species initially formed upon breaking of the C–C(HO) and C–H (aldehyde) bonds, in this medium. Recombination of the radicals formed in the matrix cage originally occupied by the BPCA molecule is a very probable event, justifying the reduced efficiency of the observed decarbonylation process, in particular in the matrices made by the smaller Ar and Kr host atoms, which can better accommodate the guest molecules and generate smaller trapping cages.

Prompt recombination of radical species can also justify the fact that the Br atom extrusion reaction was not observed for the matrix-isolated compound. However, in gas phase, according to the mass spectrum of BPCA [15], which is presented and duly assigned in Appendix A, the major species resulting from the fragmentation of the BPCA molecule are the [C_5_NH_3_Br+H]^+^ and [C_5_NH_3_+H]^+^ ions, which are compatible with the occurrence of decarbonylation and decarbonylation+debromination reactions, respectively, upon electron ionization (which shall then also produce the CO, H_2_, H_2_CO and HBr neutral species, which are not detectable in the mass spectra). Observation of the [CHO]^+^ ion in the mass spectra of BPCA (see Appendix A) is an additional proof of the relevance of the decarbonylation reaction, while the observation of the [C_4_H_3_]^+^ and [C_4_H_3_–H]^+^ ions indicates that the fragmentation of the pyridine ring upon electron ionization also takes place and shall also lead to production of HCN and ethylene. Naturally, under matrix-isolation conditions and upon the less energetic UV irradiation (compared to electron bombardment at ~70 eV used in the mass spectrometry experiments), the pyridine fragmentation cannot take place.

The performed UV irradiation was made using broadband radiation with λ ≥ 235 nm (see Section 2). In the UV, BPCA shows absorption bands with maxima at 283.4/276.8 and 238.4 nm (in ethanol solution; Figure 4), which according to the performed TD-DFT calculations, shall be assigned mostly to the HOMO→LUMO and HOMO→LUMO+1 transitions, respectively (Table 5). The performed irradiations led to excitation to the *S*_2_ to *S*_5_ states and can be expected to allow for facile intersystem crossing to the triplet manifold, considering the existence of nine triplet states with energy below *S*_5_, where the bond cleavages leading to formation of the intermediate radical species involved in the decarbonylation of BPCA take place [39]. On the other hand, taking into account previous results for other aromatic aldehydes [4,5,6,7,8,38], the observed photo-induced *trans*→*cis* BPCA conformational isomerization occurs, much probably, in the singlet manifold (either in *S*_1_ or *S*_0_, after internal conversion).

### 4.4. Room Temperature IR and Raman Spectra of Crystalline BPCA and Intermolecular Interactions in the Crystal

As mentioned in the Introduction (Section 1), the structure of the BPCA crystal has already been reported [14]. The crystal belongs to the *P*2_1_/*a* (no. 14) point group, which exhibits an inversion center, so that its IR and Raman spectra must obey the rule of mutual exclusion. These spectra (obtained for the powdered neat compound) are shown in Figure 5, where they are compared with the corresponding calculated spectra for the *trans* conformer (the form existing in the crystal of the compound) [14]. The infrared spectrum of the matrix-isolated *trans* conformer of BPCA (Xe matrix data; already presented in Figure 2) is also included in the figure for a direct comparison with the spectrum of the neat solid BPCA.

As sown in Figure 5, both IR and Raman spectra of crystalline BPCA are very well reproduced by the corresponding calculated spectra of the isolated *trans* conformer molecule. In addition, the infrared spectrum of the crystalline phase (which matches the one included in SpectraBase, spectrum ID 2XLMvrNG6Oo) [15] is very similar to those of the matrix-isolated molecules of the compound. These results demonstrate that the intermolecular interactions existing in the room temperature crystal of BPCA do not significantly perturb the intramolecular vibrational potential of the molecule.

The assignment of the spectra of crystalline BPCA is given in Table 6 and did not pose any particular difficulty, considering their resemblance to those predicted theoretically (and to the matrix spectra). As all other spectral regions analyzed, the region below 400 cm^−1^ of the Raman spectrum is also very well reproduced by the spectrum calculated for the *trans* conformer of BPCA, the predicted bands at 297, 271, 225, 163, 151 and 95 cm^−1^ having experimental counterparts observed at 301, 276, 229, 181, 167 and 79 cm^−1^, all of them showing a good match also regarding intensities (see Figure 5). Bands due to intermolecular modes are observed at 138, 93 and ~54 cm^−1^. The band at 93 cm^−1^ appears at a frequency very close to that predicted for the aldehyde torsional vibration, but its relative intensity is too low compared to that predicted for this mode by the calculations, which in turn is well described by the intensity of the band observed at 79 cm^−1^, thus justifying the proposed assignments for these two bands.

It can be seen in Table 6 that the frequencies of the bands observed in the IR and Raman spectra are very similar, i.e., for most of the vibrations, the Davydov splitting (factor group splitting) is very small (≤2 cm^−1^), with a maximum value of 5 cm^−1^ for the ring torsional mode (τring3), which implies a considerable movement of both the N atom and of the C–H bond in position *para*. This observation is in agreement with the conclusion, extracted from the comparison of both the structural parameters and vibrational spectra observed for the crystalline material with those of the isolated BPCA molecule (Section 4.1 and the paragraph above), that the intermolecular interactions in the BPCA crystal do not significantly perturb the intramolecular potential.

In order to better characterize the dominant intermolecular interactions in the crystal of the studied compound, the Hirshfeld surface analysis method [29,30,31] was used. Maps of the normalized contact distances, *d_norm_*, to the calculated Hirshfeld surface (plotted on the surface itself) and the respective 2D-fingerprint plots were obtained and are shown in Figure 6. The normalized contact distances (*d_norm_*) are calculated from the distances of a given point of the surface to the nearest atom outside, *d_e_*, and inside, *d_i_*, the surface, as $d_{norm}=\frac{d_{i}-r_{i}^{vdW}}{r_{i}^{vdW}}+\frac{d_{e}-r_{e}^{vdW}}{r_{e}^{vdW}}$ (where *r_i_^vdW^* are the van der Waals radii), and they allow for the identification of the most important intermolecular contacts in the crystal [29,30,31]. The 2D-fingerprint plots are used to condense the information about the different types of contacts and enable to access their relative importance [29,30,31]. In Figure 6, the percentages of the Hirshfeld surfaces assigned to the different types of contacts (*d_e_* vs. *d_i_*), which are shown graphically in the 2D-fingerprint plots, are also given.

As shown in the 2D-fingerprint maps depicted in Figure 6, the shortest contacts between neighbor molecules in the BPCA crystal are longer than 2.4 Å, which means that they are not particularly short. This is in agreement with the above conclusions that the intermolecular interactions in the BPCA crystal are not strong. The strongest intermolecular interactions, which correspond to the peaks appearing in the left-down quarter of the fingerprint maps, are: (*i*) the H^…^N/N^...^H contact that links the BPCA molecules into chains along the direction of the *b* axis of the crystal via C–H^...^N hydrogen-bond-like interactions (d_H…N_ = 2.53(6) Å; d_C…N_ = 3.534(7) Å; ∠_C–H…N_ = 167(4)°), where the *para* C4–H bond of the pyridine ring acts as donor (see also the *d_norm_* map on the Hirshfeld surface corresponding to the H^…^N/N^...^H interactions shown in Figure 7, where the region corresponding to this short contacts is highlighted), (*ii*) the bifurcated H^…^O/O^...^H contacts (better represented as (H′)(H″)^…^O/(O′)(O″′)^...^H contacts, where the plicas distinguish different neighboring molecules) formed between the aldehyde oxygen atoms and the *meta* H7 hydrogen atoms (in position 3 of the pyridine ring) of neighboring molecules belonging to vicinal columns of BPCA units that are constructed along the *a* axis direction (see Appendix A) (d_H…O_ = 2.64(6) Å; d_C…O_ = 3.482(7) Å; ∠_C–H…O_ = 138(4)°), and (*iii*) the H^…^H contacts between the stacked BPCA molecules in the columns, established specifically between the *para* hydrogen (H8) atoms and between the *meta* (H7) and aldehyde hydrogen atoms of adjacent molecules (Appendix A). The whole set of H^…^N/N^...^H, H^…^O/O^...^H and H^…^H interactions is associated with 40% of the total surface area of the *d_norm_* map on the Hirshfeld surface (6.2%, 19.2% and 15.2%, respectively).

Also relevant, comprehending 27.7% of the total surface area of the *d_norm_* map on the Hirshfeld surface, are the H^…^Br/Br^...^H contacts, in which the pyridine ring *meta* H9 hydrogen atoms participate (see Figure 6). These interactions are established between neighboring molecules belonging to vicinal columns of BPCA molecules defined along the *a* axis direction (d_H…Br_ = 3.24(6) Å; d_C…Br_ = 4.033(6) Å; ∠_C–H…N_ = 148(5)°), in the inter-columns’ connecting regions that appeared alternating with the ones defined by the aldehyde moieties (see Appendix A). In these bromine-rich regions of the supramolecular architecture of the crystal, Br^…^Br contacts are also relevant and contribute to 5.8% of the total surface area of the *d_norm_* map on the Hirshfeld surface, as shown in Figure 6.

In total, the most relevant contacts connecting the BPCA molecules in the crystal in the chains along the *b* axis direction (H^…^N/N^...^H) and those present in the two alternate distinct regions joining the columns of molecules formed along the *a* axis direction (H^…^O/O^...^H and H^…^H in the one side, and H^…^Br/Br^...^H and Br^…^Br in the other side) account for 74.1% of the *d_norm_* map on the Hirshfeld surface, clearly revealing the major role of these interactions in defining the structure of the BPCA crystal. The essentially polarizability-dependent H^…^Br and Br^…^Br interactions play an important role in defining the BPCA crystal structure, and this role is similar to that of the mostly dipolarity-dependent H^…^O/O^…^H interactions in keeping the stacked columns of BPCA molecules held together. The remaining fraction (25.9%) of the *d_norm_* map on the Hirshfeld surface is mainly associated with C^…^C (7.3%), H^…^C/C^…^H (6.6%) and C^…^N/N^…^C (6.5%) contacts, which are related with the staking of the BPCA molecules in the columns.

To further evaluate the strength of the whole set of intermolecular interactions in crystalline BPCA, it is interesting to compare the melting point (at 1 atm.) of the compound with those of related compounds. Along the F, Cl, Br series of 6-halogeno-pyridine-2-carbaldehydes, the fluoro-substituted compound is a liquid at normal pressure (as they are pyridine, 2-bromopyridine and pyridine-2-carbaldehyde, whose melting points are –41.6, −40.1 and −21 °C, respectively) [40], while both the chloro- and bromo-substituted derivatives are solids with melting points of 69–70 and 77–78 °C, respectively [41]. These data indicate that the presence of the more polarizable halogen substituent at position-6 of the pyridine ring leads to an increase in the strength of the intermolecular interactions in the crystals of the 6-halogeno-pyridine-2-carbaldehydes, in agreement with the Hirshfeld analysis results presented above for BPCA, which reveal the relevance of the Br-polarizability-associated interactions to stabilize the crystal structure of the compound. The effect of the halogen substitution in the pyridine ring on the melting point is synergic with the simultaneous presence of the aldehyde substituent at position-2 of the ring, since the sole presence of either the bromo or aldehyde substituent in the *ortho* position of the pyridine ring does not appreciably change the melting point. Nevertheless, pyridine-2-carbaldehyde has a melting point higher than 2-bromopyridine, an observation that is in agreement with the Hirshfeld analysis performed on the crystal of BPCA. In fact, while as mentioned above both the H^…^Br and Br^…^Br interactions and the H^…^O interactions in the crystal of BPCA play similar structural roles in keeping the stacked columns of BPCA molecules held together, the Hirshfeld analysis revealed that the latter interactions are considerably stronger.

## 5. Conclusions

In this study, 6-bromopyridine-2-carbaldehyde was studied both in the room temperature crystalline phase and isolated in cryogenic matrices (Ar, Kr and Xe). The main experimental technique used was infrared spectroscopy, while theoretical calculations at the DFT(B3LYP) level of approximation were also performed.

In the low-temperature matrices prepared from the room temperature gaseous compound, the isolated molecules of BPCA were shown to exist in a single conformer, *trans*, which could be characterized in detail vibrationally. This conformer (which is also the form present in the room temperature crystal of the compound) is stabilized by C–H^…^O=C and C(=O)H^…^N interactions that have an essentially bond–dipole/bond–dipole character. Broadband UV irradiation of the matrix-isolated BPCA (λ ≥ 235 nm) resulted in the conversion of the *trans* conformer into the higher-energy *cis* conformer, where repulsive C–H^…^H–C(=O) and C=O_LP_^…^_LP_N (where LP designates a lone electron pair) interactions are present, as well as decarbonylation of the compound with formation of 2-bromopyridine (and CO) (Figure 7). The generation of the higher-energy conformer of BPCA also allowed for its full vibrational characterization. The decarbonylation reaction was found to be more efficient in the more polarizable Xe matrix, indicating stabilization of the radicals initially formed upon breaking of the C–C(HO) and C–H (aldehyde) bonds in this medium and testifying the occurrence of the decarbonylation reaction with involvement of radical species.

TD-DFT calculations were used to access the nature of the excited states associated with the observed UV-induced reactions, showing that excitation takes place to the *S*_2_–*S*_5_ states. Subsequently, the molecule can undergo intersystem crossing to the triplet manifold (the TD-DFT calculations revealed that there are nine triplet states with energy below *S*_5_), where the bond cleavages leading to formation of the intermediate radical species involved in the decarbonylation of BPCA take place. On the other hand, the photo-induced *trans*→*cis* BPCA conformational isomerization occurs, much probably, in the singlet manifold (either in *S*_1_ or *S*_0_, after internal conversion). It was also concluded that under the used matrix-isolation conditions, no additional photo-induced reactions; in particular, pyridine ring valence isomerization and Br atom extrusion take place.

**Figure 7 molecules-28-01673-f007:**
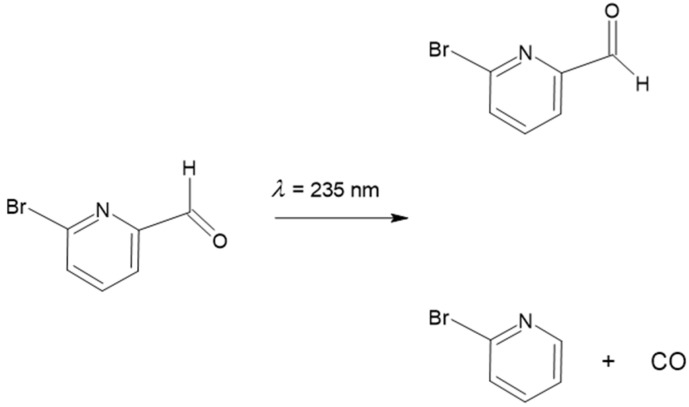
Summary of the observed photochemical processes for matrix-isolated BPCA upon irradiation at *λ* = 235 nm.

For the crystalline compound, infrared and Raman spectra were obtained and interpreted. Comparison of the obtained infrared spectrum of the crystal with those obtained for the isolated molecules of BPCA in the studied cryomatrices helped to conclude that the intermolecular interactions in the crystal do not significantly perturb the intramolecular vibrational potential. The existence of weak coupling between the intermolecular interactions and the structure of the constituting molecular units in crystalline state was also supported by the performed structural analysis and through Hirshfeld analysis. This latter revealed that the most important interactions in the BPCA crystal are weak and of the H^…^N, H^…^O, H^…^H, H^…^Br and Br^…^Br types, and indicated that the essentially polarizability-dependent H^…^Br and Br^…^Br interactions in the crystal of BPCA play a relevant role in stabilizing the crystal, which is similar to that of the mostly dipolarity-dependent H^…^O/O^…^H interactions, and that is mostly to ensure the stacked columns of BPCA molecules (formed along the direction of the crystallographic axis (*a*)) held together. The Hirshfeld analysis also indicated that, even though the H^…^Br and Br^…^Br interactions and the H^…^O interactions play similar structural roles in the crystal of BPCA, the latter interactions are considerably stronger, a result that is in consonance with the melting point data for the series of 6-halogeno-pyridine-2-carbaldehydes and the parent related compounds, pyridine, 2-bromopyridine and pyridine-2-carbaldehyde.

As a whole, this study provided fundamental data to understand the physicochemical behavior of the compound, bridging the properties of the isolated molecule to those of the neat crystal-line compound. Such information is of fundamental importance for the understanding of the role of BPCA in supramolecular chemistry and to potentiate its applications in synthesis and as a ligand for transition metal catalysts and luminescent complexes.

## Figures and Tables

**Figure 1 molecules-28-01673-f001:**
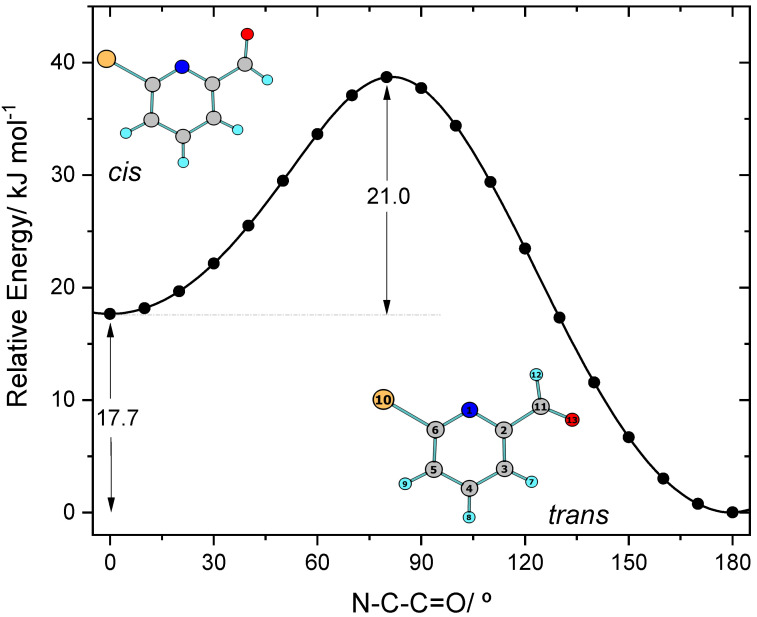
Conformers of BPCA and potential energy profile for their interconversion through rotation about the exocyclic C–C bond, as predicted at the DFT(B3LYP)/6-311++G(d,p) level of theory. Atom numbering is indicated. Atom color code: C, grey; O, red; N, blue; Br, orange; H, cyan.

**Figure 2 molecules-28-01673-f002:**
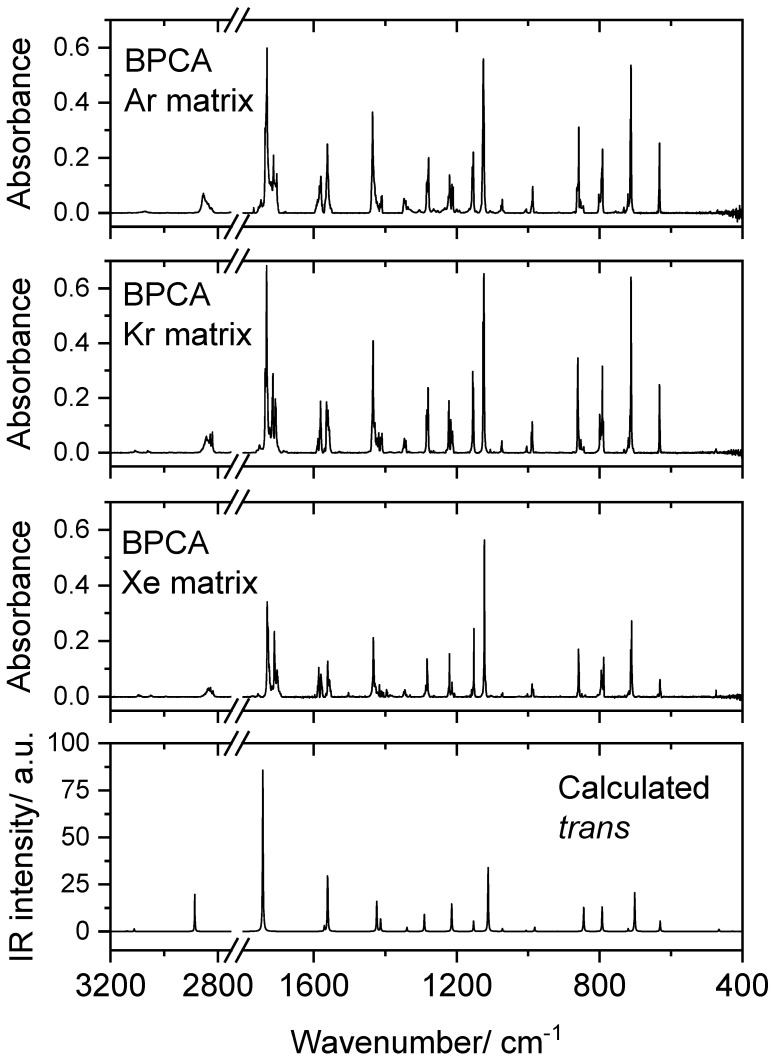
Experimental IR spectra of BPCA isolated in Ar, Kr and Xe matrices (15 K), and simulated IR spectrum of the *trans* conformer of the molecule, built using the B3LYP/6-311++G(d,p) calculated vibrational data (frequencies were scaled as described in Section 3).

**Figure 3 molecules-28-01673-f003:**
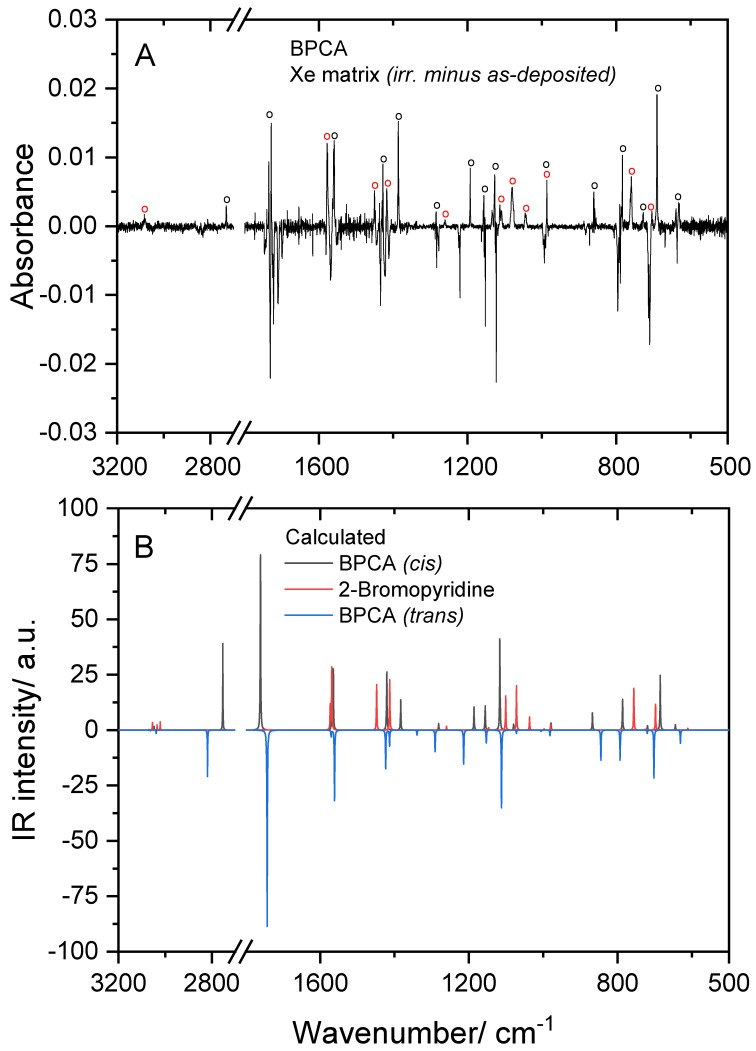
Experimental IR difference spectrum (UV-irradiated Xe matrix *minus* as-deposited matrix (**A**), and simulated IR spectrum built using the B3LYP/6-311++G(d,p) calculated vibrational data (frequencies were scaled as described in Section 3), with bands due to BPCA conformer *cis* and to 2-bromopyridine pointing up and bands due to BPCA conformer *trans* pointing down (**B**). The irradiation was performed at *λ* ≥ 235 nm for 180 min.

**Figure 4 molecules-28-01673-f004:**
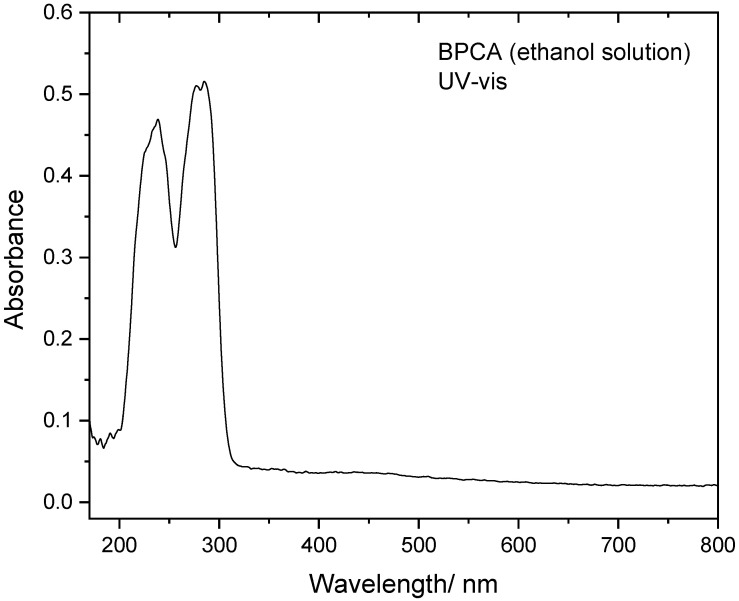
Room temperature UV–vis absorption spectrum of BPCA in ethanol solution (0.7 mM).

**Figure 5 molecules-28-01673-f005:**
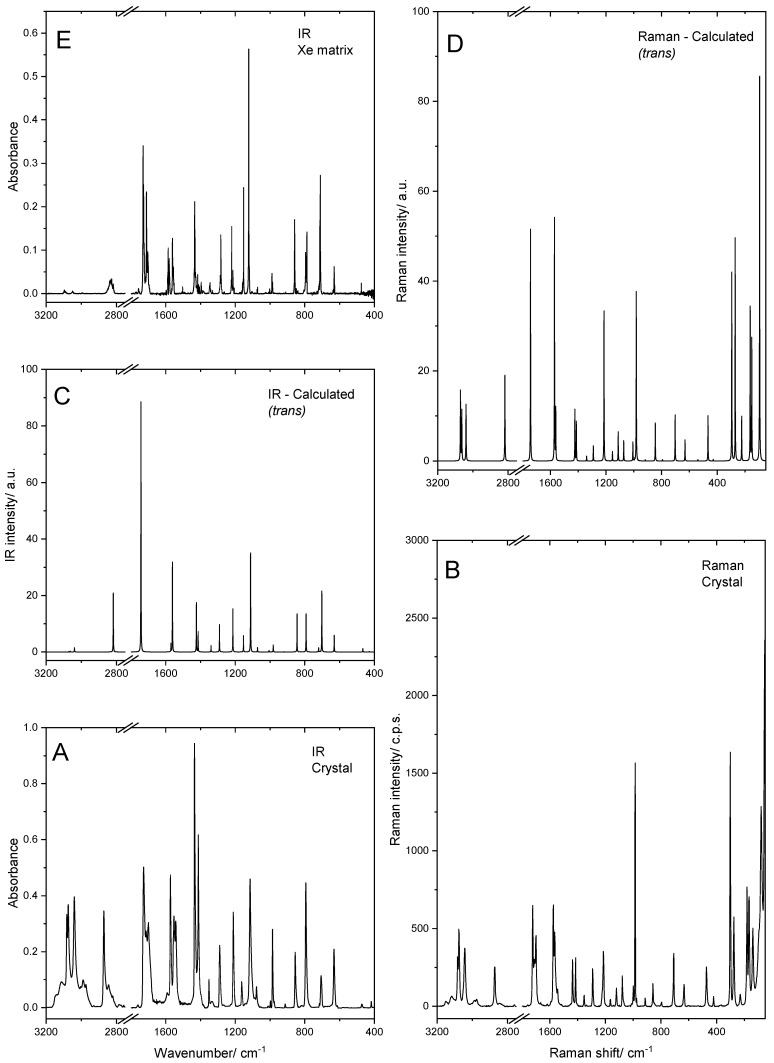
Experimental room temperature IR (**A**) and Raman (**B**) spectra of crystalline neat BPCA, compared to the corresponding simulated spectra built based on the DFT(B3LYP)/6-311++G(d,p) data (**C**,**D**). The spectrum of the isolated compound in the Xe matrix at 15 K is also shown for comparison (**E**). The calculated frequencies were scaled as described in Section 3.

**Figure 6 molecules-28-01673-f006:**
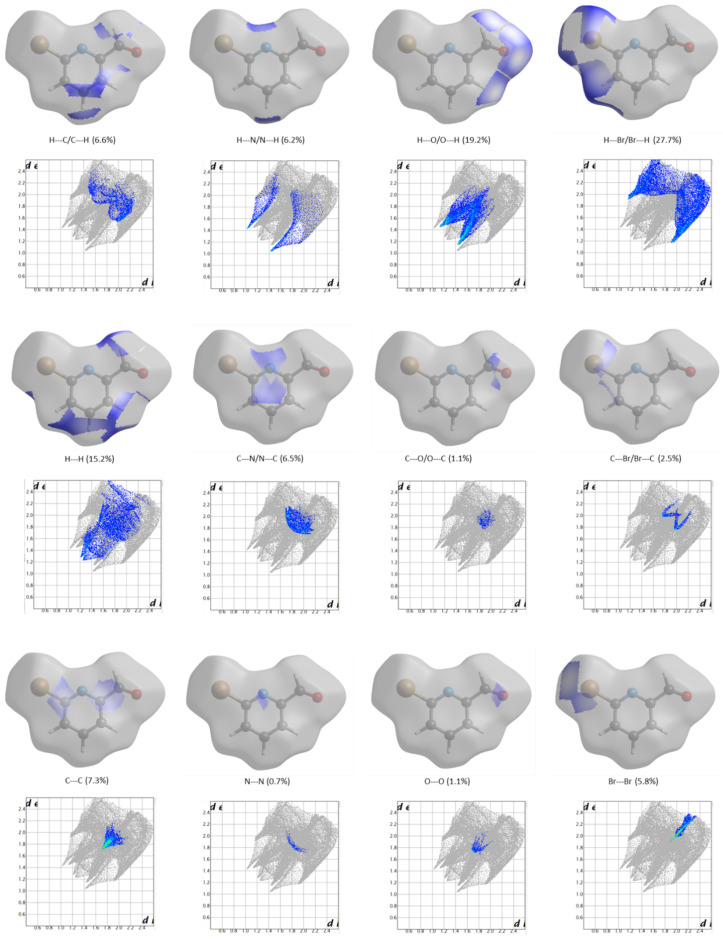
Three-dimensional *d*_norm_ maps on the Hirshfeld surface of the asymmetric unit cell of crystalline BPCA [14] and the corresponding 2D fingerprint plots (*d*_e_ vs. *d*_i_), showing the most relevant types of short contacts present in the crystal. Fractional areas (in %) of the Hirshfeld surface assigned to the different intermolecular contacts are also provided in the figure, corresponding to the blue areas in the pictures.

**Table 1 molecules-28-01673-t001:** B3LYP/6-311++G(d,p) calculated selected geometrical parameters for the two conformers of BPCA and the corresponding observed values in the crystal of the compound *^a^*.

Parameter	*cis*	*trans*	Crystal (XRD) ^14, *b*^ *(trans)*	Parameter	*cis*	*trans*	Crystal (XRD) ^14^ *(trans)*
C–Br	1.925	1.924	1.901 (5)	C2–N–C6	117.7	117.7	117.1 (4)
C=O	1.203	1.208	1.207 (6)	N–C6–C5	124.8	124.4	124.6 (4)
C–H12	1.111	1.106	1.05 (6)	N–C2–C3	122.7	123.1	122.7 (4)
C2–N	1.345	1.345	1.341 (6)	N–C2–C11	118.1	115.3	114.8 (4)
C6–N	1.307	1.312	1.296 (6)	C3–C2–C11	119.2	121.6	122.5 (4)
C2–C11	1.498	1.494	1.481 (7)	C–C=O	125.3	123.7	123.6 (5)
C2–C3	1.393	1.393	1.376 (7)	C–C–H12	113.5	113.7	110 (3)
C3–H7	1.084	1.082	1.04 (4)	O=C–H	121.2	122.5	127 (3)
C3–C4	1.396	1.392	1.364 (7)	C2–C3–H7	120.3	119.5	122 (3)
C4–C5	1.386	1.391	1.368 (7)	N–C–Br	117.1	117.0	116.4 (4)
C5–C6	1.403	1.399	1.390 (8)	C5–C6–Br	118.1	118.6	119.0 (4)
				N–C–C=O	0.0	180.0	178.4

*^a^* Bond lengths in Å; angles in degrees. See Figure 1 for structures of the conformers and atom numbering. *^b^*The experimentally determined C–H bond lengths lie in the range 0.91 (5)–1.05 (6) Å.

**Table 2 molecules-28-01673-t002:** Assignment of the IR spectrum of the as-deposited BPCA isolated in Ar, Kr and Xe matrices (conformer *trans*) and calculated PEDs *^a^*.

Ar Matrix	Kr Matrix	Xe Matrix	Calculated			Assignment (PED) *^c^*
$\tilde{\nu }$	$\tilde{\nu }$	$\tilde{\nu }$	$\tilde{\nu }$	I^IR^		
3130	3108, 3100	3096, 3088	3070.6	0.4	A′	νC4–H (91)
3130	3108, 3100	3096, 3088	3063.1	1.0	A′	νC2/3–H s (68) + νC2/3–H a (28)
3075, 3069	3061.3055	3052, 3048	3038.4	4.5	A′	νC2/3–H a (72) + νC2/3–H s (24)
2856, 2853, 2852, 2845, 2836, 2824 *^b^*	2847, 2843, 2839, 2829, 2821 *^b^*	2847, 2842, 2838, 2833, 2828, 2818 *^b^*	2818.3	63.9	A′	νC–H_al_ (100)
1736, 1733, 1731, 1728, 1727, 1725, 1723, 1720, 1718, 1713, 1708, 1704 *^b^*	1736, 1732, 1729, 1723, 1718, 1716, 1714, 1708, 1706 *^b^*	1730, 1728, 1725, 1724, 1722, 1717, 1710, 1709, 1703, 1701 *^b^*	1751.4	269.6	A′	νC=O (91)
1592, 1590	1591, 1588	1589, 1586	1578.5	8.7	A′	νring1 (56) + δCH_r_ 3 (15) + νring2 (13)
1587, 1584, 1580, 1568, 1565, 1562, 1560, 1556 *^b^*	1584, 1581, 1570, 1564, 1561, 1558, 1557 *^b^*	1580, 1577, 1566, 1560, 1557, 1552 *^b^*	1569.3	95.8	A′	νring2 (61) + δCH_r_ 2 (11)
1437, 1435, 1432, 1430, 1427, 1423, 1420, 1418	1438, 1434, 1430, 1426, 1423, 1420, 1418	1436, 1434, 1432, 1429, 1427, 1423, 1420, 1416	1431.0	52.5	A′	δCH_r_ 1 (54) + νring4 (30)
1415, 1411, 1409	1413, 1411, 1408	1411, 1410, 1405	1420.5	21.5	A′	νring5 (37) + δCH_r_ 3 (36) + δCH_al_ (17)
1349, 1348, 1345, 1343	1351, 1347, 1346, 1342	1348, 1347, 1345, 1340	1346.3	6.9	A′	δCH_al_ (82)
1284, 1281, 1279	1285, 1283, 1280	1290, 1286, 1283, 1282	1297.3	29.4	A′	νring6 (66)
1223, 1220, 1219, 1213, 1211 *^b^*	1223, 1221, 1216, 1215, 1212, 1210 *^b^*	1223, 1221, 1216, 1213, 1211, 1206 *^b^*	1220.1	46.1	A′	δCH_r_ 1 (33) + νC–C (24) + νring6 (14) + νring4 (13)
1156, 1153	1155, 1153	1154, 1152	1158.8	17.5	A′	δCH_r_ 2 (61) + νring6 (15)
1128, 1125	1126, 1124	1123, 1120	1117.8	108.1	A′	νring4 (33) + νring3 (16) + δring1 (14) + δCH_r_ 2 (12)
1076, 1073	1075, 1074	1074, 1072	1077.5	4.7	A′	δCH_r_ 3 (43) + νring5 (40)
1008, 1005	1005, 1004	1007, 1002	1011.0	1.7	A″	γCH_al_ (71) + γCH_r_ 2 (20)
n.obs.	n.obs.	993	997.7	0.03	A″	γCH_r_ 2 (87) + γCH_al_ (16)
990, 989, 987	991, 988	989, 988, 986	986.8	7.4	A′	δring1 (53) + νring3 (43)
916, 913	913, 910	910	922.4	0.2	A″	γCH_r_ 3 (100)
863, 861, 859, 855, 853, 849, 846	861, 858, 852, 844	862, 859, 857, 850, 841, 839	849.1	41.7	A′	νC–C (26) + δring3 (18) + νring4 (13) + νring3 (10)
802, 800, 795, 793, 792 *^b^*	799, 796, 793, 790 *^b^*	798, 795, 792, 789 *^b^*	797.1	40.9	A″	γCH_r_ 1 (84)
732, 725, 721	731, 723, 719	730, 723, 720, 717	723.7	4.4	A″	τring2 (94)
714, 712	715, 712	713, 711	705.6	64.8	A′	δring3 (27) + δC=O (27) + νC–Br (13)
633	632	631, 629	634.0	18.1	A′	δring2 (64) + δC=O (19)
n.obs.	n.obs.	n.obs.	541.5	0.002	A″	γCBr (37) + τring1 (32) + γCHO (29)
470	474	475	468.3	3.8	A′	δring3 (37) + νC–C (25) + wCHO (13)
n.obs.	n.obs.	n.obs.	431.0	0.6	A″	τring3 (91)
n.i.	n.i.	n.i.	296.7	8.6	A′	νC–Br (54) + δCBr (15)
n.i.	n.i.	n.i.	271.2	6.6	A′	δCBr (33) + δC=O (22) + wCHO (16) + νC–Br (14)
n.i.	n.i.	n.i.	225.0	15.9	A″	τC–CHO (38) + γCHO (31) + γCBr (14) + τring1 (12)
n.i.	n.i.	n.i.	163.2	0.8	A″	τring1 (65) + γCHO (25)
n.i.	n.i.	n.i.	151.9	2.9	A′	wCHO (43) + δCBr (38)
n.i.	n.i.	n.i.	94.9	3.6	A″	τC–CHO (52) + γCHO (17) + τring3 (13)

*^a^* Frequencies in cm^−1^; calculated intensities (I^IR^) in km mol^−1^; calculated frequencies scaled by 0.983 and 0.955, below and above 1800 cm^−1^, respectively; n.obs., not observed; n.i., not investigated. *^b^* Fermi resonance (see text for discussion). *^c^* ν, stretching; δ, in-plane bending; γ, out-of-plane rocking; w, wagging; τ, torsion; al, aldehyde; r, ring; s. symmetric; a, anti-symmetric. PEDs are expressed in %, and the PED values lower than 10% are not included. Definition of internal coordinates is given in Appendix A.

**Table 3 molecules-28-01673-t003:** Assignment of the IR spectrum of the photoproduced *cis* conformer of BPCA in Ar, Kr and Xe matrices and calculated PEDs *^a^*.

Ar Matrix	Kr Matrix	Xe Matrix	Calculated			Assignment (PED) *^c^*
$\tilde{\nu }$	$\tilde{\nu }$	$\tilde{\nu }$	$\tilde{\nu }$	I^IR^		
n.obs	n.obs	n.obs	3071.0	0.3	A′	νC4–H (94)
n.obs.	n.obs.	n.obs.	3047.0	5.0	A′	νC2/3–H s (95)
n.obs.	n.obs.	n.obs.	3034.0	2.8	A′	νC2/3–H a (98)
2744	2730	2732	2752.9	117.6	A′	νC–H_al_ (100)
1759, 1727 *^b^*	1745, 1734 *^b^*	1734, 1728 *^b^*	1769.6	242.3	A′	νC=O (92)
n.obs.	n.obs.	n.obs.	1577.8	16.9	A′	νring1 (65) + δCH_r_ 3 (18)
1573	1566	1560, 1558	1572.5	82.9	A′	νring2 (71) + δCH_r_ 2 (13)
1434	1430	1430	1429.7	5.1	A′	νring5 (30) + δCH_r_ 3 (19) + δCH_al_ (16)
1434	1430	1427	1427.9	79.2	A′	δCH_r_ 1 (47) + νring4 (24)
1389	1389, 1387	1386	1390.4	41.8	A′	δCH_al_ (84)
1288	n.obs.	1285	1287.8	9.1	A′	νring6 (77)
n.obs.	1195	1193	1192.0	31.3	A′	νC–C (24) + δCH_r_ 1 (18) + νring4 (14) + δring1 (14) + νring3 (13)
1160	1158, 1157	1157, 1156	1161.9	33.4	A′	δCH_r_ 2 (59) + νring6 (15)
1131, 1129	1129, 1127	1127	1122.3	123.5	A′	νring4 (40) + δCH_r_ 1 (12) + δCH_r_ 2 (12)
n.obs.	n.obs.	1089	1085.1	8.1	A′	νring5 (40) + δCH_r_ 3 (38) + νring3 (11)
n.obs.	n.obs.	n.obs.	1002.7	1.5	A″	γCH_al_ (81)
n.obs.	n.obs.	n.obs.	991.4	0.003	A″	γCH_r_ 2 (100)
986	985	986	984.0	9.5	A′	δring1 (57) + νring3 (39)
n.obs.	n.obs.	n.obs.	909.1	0.1	A″	γCH_r_ 3 (100)
860, 852, 848	867	860, 857, 854	871.9	23.8	A′	νC–C (28) + δC=O (15) + δring3 (12)
804, 798	788, 785	790, 783	790.4	41.8	A″	γCH_r_ 1 (82)
n.obs.	719	727	723.1	6.0	A″	τring2 (90)
692	691	690	688.7	74.9	A′	δring3 (45) + δC=O (30)
n.obs.	647	630	647.7	7.2	A′	δring2 (63)
n.obs.	n.obs.	n.obs.	545.2	0.2	A″	γCBr (37) + γCHO (31) + τring1 (30)
n.obs.	n.obs.	n.obs.	435.5	0.001	A″	τring3 (84)
445	432	434	435.5	1.9	A′	νC–C (29) + δring3 (20) + δring2 (14) + wCHO (10)
n.i.	n.i.	n.i.	309.8	10.9	A′	νC–Br (35) + δCBr (25) + wCHO (13) + δC=O (13)
n.i.	n.i.	n.i.	292.7	2.6	A′	νC–Br (36) + δCBr (31)
n.i.	n.i.	n.i.	218.5	8.0	A″	τring1 (40) + τC–CHO (28) + γCHO + τring3 (12)
n.i.	n.i.	n.i.	153.5	2.6	A″	γCBr (44) + τring1 (19) + γCHO (13) + τring3 (12)
n.i.	n.i.	n.i.	136.5	3.0	A′	wCHO (43) + δCBr (34) + δC=O_l_ (12)
n.i.	n.i.	n.i.	90.2	3.5	A″	τC–CHO (64) + τring1 (20) + γCHO (13)

*^a^* Frequencies in cm^−1^; calculated intensities (I^IR^) in km mol^−1^; calculated frequencies scaled by 0.983 and 0.955, below and above 1800 cm^−1^, respectively; n.obs., not observed; n.i., not investigated. *^b^* Fermi resonance. *^c^* ν, stretching; δ, in-plane bending; γ, out-of-plane rocking; w, wagging; τ, torsion; al, aldehyde; r, ring; s. symmetric; a, anti-symmetric. PEDs are expressed in %, and PED values lower than 10% are not included. Definition of internal coordinates is given in Appendix A.

**Table 4 molecules-28-01673-t004:** Assignment of the IR spectrum of the photoproduced 2-bromopyridine in Ar, Kr and Xe matrices and calculated PEDs *^a^*.

Ar Matrix	Kr Matrix	Xe Matrix	Calculated			Assignment (PED) *^b^*
$\tilde{\nu }$	$\tilde{\nu }$	$\tilde{\nu }$	$\tilde{\nu }$	I^IR^		
n.obs.	n.obs.	n.obs.	3069.3	0.8	A′	νCH_r_ 1 (48) + νCH_r_ 2 (33)
3098	3088	3082	3053.9	10.3	A′	νCH_r_ 1 (42) + νCH_r_ 4 (36) + νCH_r_ 2 (22)
3080	n.obs.	n.obs.	3034.2	7.0	A′	νCH_r_ 3 (47) + νCH_r_ 2 (33) + νCH_r_ 4 (20)
3080	n.obs.	n.obs.	3020.9	11.2	A′	νCH_r_ 3 (44) + νCH_r_ 4 (35) + νCH_r_ 2 (11)
1580	n.obs.	1577	1580.9	33.9	A′	νring1 (60) + νring2 (14)
1580	n.obs.	1577	1576.7	86.2	A′	νring2 (57) + δCH_r_ 4 (13) + νring1 (10)
1452, 1440	1462	1451	1455.4	61.8	A′	δCH_r_ 2 (50) + νring5 (27) + δCH_r_ 3 (11)
1427, 1423, 1416	1417	1417	1420.0	69.3	A′	δCHr 1 (58) + νring4 (25)
n.obs.	n.obs.	n.obs.	1290.6	0.9	A′	δCH_r_ 4 (35) + δCH_r_ 1 (27) + νring1 (14) + νring5 (11)
n.obs.	1270	1260	1266.5	5.4	A′	νring6 (80) + δCH_r_ 3 (10)
n.obs.	n.obs.	1133	1152.8	3.2	A′	δCH_r_ 3 (70) + νring6 (16)
1113, 1097	1113	1113, 1109	1106.5	46.4	A′	νring4 (42) + δCH_r_ 4 (23) + δCHr 1 (11)
1082	1082	1080	1077.3	60.6	A′	νring5 (29) + δring 1 (21) + δCH_r_ 2 (16) + δCH_r_ 4 (15)
1046	1051	1044	1042.0	17.8	A′	νring3 (37) + νring4 (16) + νring5 (15) + ring 1 (15)
n.obs.	n.obs.	n.obs.	988.5	0.001	A″	γCH_r_ 3 (100)
986	985	986	985.0	7.4	A′	δring 1 (54) + νring3 (42)
n.obs.	n.obs.	n.obs.	961.9	0.3	A″	γCH_r_ 4 (95)
n.obs.	n.obs.	n.obs.	881.2	0.01	A″	γCH_r_ 2 (88)
768	770	760	760.0	57.2	A″	γCH_r_ 1 (85)
n.obs.	n.obs.	n.obs.	729.6	1.0	A″	τring2 (100)
704	703	703	701.3	35.9	A′	δring 3 (67) + νC–Br (16)
n.obs.	n.obs.	n.obs.	614.0	2.4	A′	δring 2 (77) + δring 3 (12)
n.obs.	n.obs.	n.obs.	470.9	4.0	A″	γCBr (43) + τring1 (40) + τring3 (13)
n.i.	n.i.	n.i.	408.3	4.7	A″	τring3 (89) + τring1 (19)
n.i.	n.i.	n.i.	304.2	6.3	A′	νC–Br (74) + δring 3 (13)
n.i.	n.i.	n.i.	255.2	1.9	A′	δCBr (91)
n.i.	n.i.	n.i.	154.2	1.3	A″	τring1 (50) + γCBr (41)

*^a^* Frequencies in cm^−1^; calculated intensities (I^IR^) in km mol^−1^; calculated frequencies scaled by 0.983 and 0.955, below and above 1800 cm^−1^, respectively; n.obs., not observed; n.i., not investigated. *^b^* ν, stretching; δ, in-plane bending; γ, out-of-plane rocking; τ, torsion. PEDs are expressed in %, and the PED values lower than 10% are not included. Definition of internal coordinates is given in Appendix A.

**Table 5 molecules-28-01673-t005:** Vertical excitation energies, wavelengths, oscillator strengths and main contributions to the excited states of BPCA (*trans* conformer) calculated at the TD-DFT(CAM-B3LYP)6-311+G(d,p) level in ethanol.

Excited State	Energy (eV)	*λ*_calc._*^a^* (nm)	*f ^a^*	Main Contributions (%) *^b^*
*T* _1 (A′)_	3.17	391		HOMO→LUMO (55%) + HOMO-3→LUMO (30%) + HOMO→LUMO+1 (17%)
*T* _2 (A″)_	3.24	383		HOMO-1→LUMO (64%)
*S* _1 (A″)_	3.81	326	0.000	HOMO-1→LUMO (65%))
*T* _3 (A′)_	4.20	295		HOMO→LUMO+1 (47%) + HOMO-3→LUMO (23%)
*T* _4 (A″)_	4.40	282		HOMO-4→LUMO (43%) + HOMO-2→LUMO (37%)
*S* _2 (A′)_	4.50	275	0.130	HOMO→LUMO (65%)
*T* _5 (A′)_	4.58	271		HOMO-3→LUMO (47%) + HOMO-5→LUMO (16%)
*S* _3 (A″)_	4.61	269	0.001	HOMO-4→LUMO (49%) + HOMO-2→LUMO (43%)
*T* _6 (A′)_	5.29	234		HOMO-3→LUMO+1 (53%) + HOMO-3→LUMO (29%)
*T* _7 (A″)_	5.38	231		HOMO→LUMO+3 (29%) + HOMO-1→LUMO+1 (28%)
*T* _8 (A″)_	5.43	228		HOMO→LUMO+3 (36%) + HOMO-4→LUMO+1 (23%)
*T* _9 (A′)_	5.47	227		HOMO-7→LUMO (44%) + HOMO-5→LUMO (27%)
*S* _4 (A′)_	5.51	223	0.067	HOMO→LUMO+1 (52%) + HOMO-3→LUMO (39%) + HOMO→LUMO (21%)
*S* _5 (A″)_	5.71	217	0.004	HOMO-1→LUMO+1 (45%)
*T* _10 (A′)_	5.78	215		HOMO-2→LUMO+3 (45%) + HOMO-4→LUMO+2 (18%)
*S* _6 (A″)_	5.96	208	0.000	HOMO→LUMO+3 (48%)

*^a^* Results are shown for the region up to 200 nm. *^b^* Only contributions of ≥15% are included.

**Table 6 molecules-28-01673-t006:** Assignment of the room temperature IR and Raman spectra of BPCA crystal *^a^*.

Experimental		Calculated			Assignment *^d^*
IR *^b^* $\tilde{\nu }$	Raman $\tilde{\nu }$	(*trans*) $\tilde{\nu }$	I^IR^	I^R^	
3081	3083	3070.6	0.4	488.7	νC4–H
3074	3075	3063.1	1.0	352.5	νC2/3–H s
3039	3042	3038.4	4.5	395.3	νC2/3–H a
2871, 2846 *^c^*	2872	2818.3	63.9	598.1	νC–H_al_
1726, 1712, 1701 *^c^*	1722, 1711, 1700	1751.4	269.6	1622.0	νC=O
1572	1574, 1563	1578.5	8.7	1698.9	νring1
1553, 1542	1543	1569.3	95.8	361.9	νring2
1434	1436	1431.0	52.5	357.9	δCH_r_ 1; νring4
1413	1414	1420.5	21.5	274.0	νring5; δCH_r_ 3
1351	1352	1346.3	6.9	34.1	δCH_al_
1289	1291	1297.3	29.4	107.1	νring6
1214	1214	1220.1	46.1	1050.4	δCH_r_ 1; νC–C
1163	1163	1158.8	17.5	67.1	δCH_r_ 2
1116	1120	1117.8	108.1	202.8	νring4
1078	1078	1077.5	4.7	142.5	δCH_r_ 3; νring5
1013	1011	1011.0	1.7	130.6	γCH_al_
998	999	997.7	0.03	12.4	γCH_r_ 2
985	986	986.8	7.4	1184.2	δring1; νring3
914	914	922.4	0.2	9.2	γCH_r_ 3
854	858	849.1	41.7	264.4	νC–C; δring3
794	795	797.1	40.9	10.1	γCH_r_ 1
716	n.obs.	723.7	4.4	0.1	τring2
706	708	705.6	64.8	320.5	δring3; δC=O
632	634	634.0	18.1	148.2	δring2
n.obs.	n.obs.	541.5	0.002	9.6	γCBr; τring1
471	473	468.3	3.8	316.2	δring3
417	422	431.0	0.6	11.3	τring3
n.i.	301	296.7	8.6	1322.2	νC–Br
n.i.	276	271.2	6.6	1561.5	δCBr
n.i.	229	225.0	15.9	309.7	γCHO; τC–CHO
n.i.	181	163.2	0.8	1075.8	τring1
n.i.	167	151.9	2.9	855.3	wCHO; δCBr (38)
n.i.	79	94.9	3.6	2697.8	τC–CHO

*^a^* Frequencies in cm^−1^; calculated frequencies scaled by 0.983 and 0.955, below and above 1800 cm^−1^, respectively; calculated infrared intensities (I^IR^) in km mol^−1^; calculated Raman intensities (I^R^; differential Raman backscattering cross sections) in 10^−38^ m^2^ sr^−1^); n.obs., not observed; n.i., not investigated. *^c^* Fermi resonance. *^d^* ν, stretching; δ, in-plane bending; w, wagging; γ, out-of-plane rocking; τ, torsion; al, aldehyde; r, ring. The assignments are given considering the major contributing coordinates according to the calculated PEDs for the *trans* BPCA conformer *in vacuo* (see Table 2).

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
