# Peer review of "Structure, Vibrational Spectra, and Cryogenic MatrixPhotochemistry of 6-Bromopyridine-2-carbaldehyde: From the Single Molecule of the Compound to the Neat Crystalline Material"

_molecules, 2023, doi:10.3390/molecules28041673_

Round 1

Reviewer 1 Report

The authors report a descriptive analysis of the characterization of 6-bromopyridine-2-carbaldehyde. In particular, the authors report a chemical-physical characterization by measuring the IR spectrum, the UV spectrum and a computational analysis of BPCA at room temperature in the crystalline phase and in cryogenic matrices.

Unfortunately, the manuscript is purely descriptive. The scope of the work does not highlight the importance of the study.

What is the scientific question to answer? What are the applications of their analysis?

Without these aspects, the manuscript loses interest for the vast audience of readers of MOLECULES.

Author Response

The authors report a descriptive analysis of the characterization of 6-bromopyridine-2-carbaldehyde. In particular, the authors report a chemical-physical characterization by measuring the IR spectrum, the UV spectrum and a computational analysis of BPCA at room temperature in the crystalline phase and in cryogenic matrices.

Unfortunately, the manuscript is purely descriptive. The scope of the work does not highlight the importance of the study. What is the scientific question to answer? What are the applications of their analysis?

Without these aspects, the manuscript loses interest for the vast audience of readers of MOLECULES.

Our answer: We have to disagree with the classification by the reviewer of our manuscript as a “descriptive analysis”. In fact, in the manuscript we report a very detailed analysis of the structure and vibrational spectra of the studied compound in both its crystalline room temperature phase and for its matrix-isolated molecules. In the latter case, we also present a detailed study of the photochemistry of the compound. All reported data have been discussed in detail, focusing their relevance for the understanding of the physicochemical behavior of the compound, which is of fundamental importance for the understanding of its role in supramolecular chemistry and also as a ligand for transition metal catalysts and luminescent complexes.

The relevance of the study is in fact highlighted in the first sentence of the Abstract!... However, we understood the point raised by the reviewer and we included a few sentences in the abstract, introduction and conclusion sections to highlight in a more visible way the relevance of the study, the questions answered and the relevance of the undertaken analyses.

Reviewer 2 Report

This is basic research, but this manuscript actually presented lots of data. Publication of these data in public journal media would have certain contribution to the related research fields. From this positive view point, I may suggest publication of this work in Molecules. However, the manuscript in the current style looks like data display. This impression has to be changed through revisions. Please see below.

1) The title can be modified. Measurements at room temperature are common. Instead, investigation under cryogenic conditions is attractive. Probably, more emphasis on the latter matter is advantageous for good impression. In addition, inclusion of a new conceptual term in the title often works well for innovative impression. I may suggest use an emerging conceptual term, nanoarchitectonics, in the title (as post-nanotechnology concept, see https://pubs.rsc.org/en/content/articlelanding/2021/nh/d0nh00680g). For example, the title like ... Cryogenic-Matrix Nanoarchitectonics of 6-Bromopyridine-2-Carbaldehyde: Structure, Vibrational Spectra, and Photochemistry ... may sound more innovative.

2) This manuscript provides lots of data. This organization would make this manuscript less understandable. Aids of structural illustration is necessary. For example, one conclusive figure with chemical structures can be added at the last of this manuscript.

Author Response

Reviewer #2:

This is basic research, but this manuscript actually presented lots of data. Publication of these data in public journal media would have certain contribution to the related research fields. From this positive view point, I may suggest publication of this work in Molecules. However, the manuscript in the current style looks like data display. This impression has to be changed through revisions. Please see below.

1) The title can be modified. Measurements at room temperature are common. Instead, investigation under cryogenic conditions is attractive. Probably, more emphasis on the latter matter is advantageous for good impression. In addition, inclusion of a new conceptual term in the title often works well for innovative impression. I may suggest use an emerging conceptual term, nanoarchitectonics, in the title (as post-nanotechnology concept, see https://pubs.rsc.org/en/content/articlelanding/2021 /nh/d0nh00680g). For example, the title like ... Cryogenic-Matrix Nanoarchitectonics of 6-Bromopyridine-2-Carbaldehyde: Structure, Vibrational Spectra, and Photochemistry ... may sound more innovative.

Our answer: We have to disagree with the classification of our manuscript by the reviewer as a “data display”. In fact, in the manuscript we report a very detailed analysis of the structure and vibrational spectra of the studied compound in both its crystalline room temperature phase and for its matrix-isolated molecules. In the latter case, we also present a detailed study of the photochemistry of the compound. All reported data have been discussed in detail, focusing their relevance for the understanding of the physicochemical behavior of the compound, which is also of fundamental importance for the understanding of its role in supramolecular chemistry and also as a ligand for transition metal catalysts and luminescent complexes. A few sentences were now included in the abstract, introduction and conclusion sections to highlight in a more visible way the relevance of the study, the questions answered and the relevance of the undertaken analyses.

The reference to the cryogenic matrix studies was already in the original title. We do not agree with the introduction of the “nanoarchitectonics” concept in the title. Instead, we introduced in the title the idea of the aim to bridge the properties of the isolated molecule and neat crystalline phase.

2) This manuscript provides lots of data. This organization would make this manuscript less understandable. Aids of structural illustration is necessary. For example, one conclusive figure with chemical structures can be added at the last of this manuscript.

Our answer: The manuscript is indeed extensively illustrated [it contains 6 Figures (most of them with several panels); plus 2 additional Figures in the Supporting Information]. Nevertheless, in consonance with the reviewer request, we added now a new Figure (Figure 7) showing the structures of the chemical species relevant for the observed photochemistry of the matrix-isolated compound, which is the only observation for which in the original version of the manuscript we had not including a scheme illustrating the observations (in any case, it shall be noticed that this information was exactly the one chosen to be presented as main information in the Table of Contents Graph sent with the original manuscript).

Round 2

Reviewer 1 Report

As the authors will surely know, "descriptive analysis" does not mean poorly detailed and in-depth. The term "descriptive study" means a research that reports the description of an event, a characteristic, a process without answering a scientific question. The authors in their manuscript do not define a problem or a scientific question to be solved. For example, "the in-depth analysis of the structure and vibrational spectra of the compound studied both in its crystalline phase at room temperature and for its molecules isolated from the matrix" does not answer a scientific question, but rather, it describes! Again, the "chemical-physical behavior of the compound" is still a descriptive aspect.

In my opinion the manuscript does not fall within the topics of MOLECULES, I suggest the authors to submit the manuscript elsewhere. The choice to accept the manuscript is the responsibility of the Editor.